# Association between Lower Intake of Minerals and Depressive Symptoms among Elderly Japanese Women but Not Men: Findings from Shika Study

**DOI:** 10.3390/nu11020389

**Published:** 2019-02-13

**Authors:** Thao Thi Thu Nguyen, Sakae Miyagi, Hiromasa Tsujiguchi, Yasuhiro Kambayashi, Akinori Hara, Haruki Nakamura, Keita Suzuki, Yohei Yamada, Yukari Shimizu, Hiroyuki Nakamura

**Affiliations:** Department of Environmental and Preventive Medicine, Graduate School of Medical Science, Kanazawa University, 13-1 Takara-machi, Kanazawa 920-8640, Japan; smiyagi@staff.kanazawa-u.ac.jp (S.M.); t-hiromasa@med.kanazawa-u.ac.jp (H.T.); ykamba@med.kanazawa-u.ac.jp (Y.K.); ahara@m-kanazawa.jp (A.H.); haruki_nakamura@stu.kanazawa-u.ac.jp (H.N.); keitasuzuk@yahoo.co.jp (K.S.); yamada503597@gmail.com (Y.Y.); h_zu@me.com (Y.S.); hiro-n@po.incl.ne.jp (H.N.)

**Keywords:** depressive symptoms, elderly individuals, mineral intake, gender

## Abstract

The aim of this cross-sectional study was to examine the relationship of mineral intake, including sodium, potassium, calcium, magnesium, phosphorus, iron, zinc, copper and manganese, with depressive symptoms in both genders in the Japanese elderly population. A total of 1423 participants who were older than 65 years old were recruited in this study. Mineral intake was analyzed using a validated and brief self-administered diet history questionnaire. Depressive symptoms were assessed with a short version of the Geriatric Depression Scale. A logistic regression model was applied to determine the relationship between mineral intake and depressive symptoms. The prevalence of depressive symptoms was 20%. Except for sodium and manganese, mineral intake was significantly lower in the depressive symptoms group. There was no difference of mineral intake between male participants with depressive symptoms and those without such symptoms. However, in female participants, mineral intake was significantly lower in participants with depressive symptoms compared to those without such symptoms. Potassium, calcium, magnesium, phosphorus, iron, zinc, and copper were significantly and negatively correlated with depressive symptoms among female participants, but not male participants. Our results suggest that the deficiencies in mineral intake may be related to depressive symptoms, especially in women.

## 1. Introduction

Depression is one of the most burdensome health conditions, with a prevalence that increased by 37.5% between 1990 and 2010 [1], and has become the single largest contributor to global disability. Depression has been shown to be more common among women than men. A study conducted in the United States population showed that the lifetime incidence of depression was more than 12% in men and 20% in women [2]. A recent study in Europe indicated that women accounted for 67% of the individuals with depression [3]. The rate of depression also varies by age, as it peaks in older adulthood, affecting more than 7.5% of women aged 55–74 years and 5.5% of men [4]. In people aged 65 and older, the prevalence of major depressive disorder at lifetime was 8.2% [5], and approximately 15% of community dwelling older adults had depressive symptoms [6]. Depression in the elderly contributes to the prevalence of other diseases and medical problems [7]. In Japan, the number of patients with mood disorders, including depression, increased by 2.5 times from 433,000 patients in 1996 to 1,116,000 patients in 2014. There was a lower number of women with mood disorders (416,000 patients) compared to women (700,000 patients) [8].

The mechanism of depression has not yet been fully explained. In recent years, there has been increasing attention paid to the role of nutrients, including minerals, in the development of depression. Mineral intake, including microminerals (zinc, copper, iron, etc.) and macrominerals (potassium, calcium, magnesium, and so on), has been suggested to play important roles in depression. Previous studies suggested the possible contribution of these nutrients to the development of depression. Research in US adults has shown an inverse association between total zinc, iron, and copper intake with depression [9]. A study of 1792 men and 214 women who were Japanese employees aged 19–60 years has suggested that a higher prevalence of depressive symptoms was associated with lower dietary intake of magnesium, calcium, iron, and zinc [10]. A meta-analysis also found a significant association between inadequacy in dietary zinc and iron intake and the risk of depression [11]. In particular, several studies conducted using only female participants showed a strong relationship between mineral intake and the occurrence of depression. For example, studies on women in general and those specifically focusing on pregnant women have observed a relationship between mineral deficiency and depression [12,13,14]. However, the epidemiology data that compare the relationship between mineral intake and depressive symptoms in both genders in the general population are still limited, especially in the elderly population. Therefore, this present study aims to examine the difference between the associations of mineral intake and depressive symptoms in each gender in an elderly Japanese population.

## 2. Materials and Methods

### 2.1. Study Population

A longitudinal population-based cohort study was conducted, the participants of which were the residents of Shika town, a coastal suburb town located on Noto peninsula in Ishikawa prefecture. This study has been running since 2011 and aims to describe the health status of the population in a town model and propose interventions to improve people’s health. The details of the Shika Study and the study population have been previously described [15]. 

In brief, all individuals who are older than 40 years old and live in the model districts in Shika town were invited to participate in the study. 

By 2016, the Shika Study had a total of 4120 participants. In this present cross-sectional study, we recruited 1423 individuals who were 65 years old or older. They joined between January 2015 to January 2016 and completed all sections of the questionnaire. The details of recruitment are shown in Figure 1.

Written informed consent was obtained from all participants in the survey. The Shika Study was approved by the Ethical Committee at Kanazawa University.

### 2.2. Depressive Symptoms Assessment

The shortened version of Geriatric Depression Scale (GDS) was used to assess the depressive symptoms of participants. The GDS short version consists of fifteen yes/no questions that asks about the feelings of the participant over the past week. Of these questions, ten items indicate the presence of depressive symptoms while the other five items indicated depressive symptoms when answered negatively. The total score of GDS is out of 15, with a higher score indicating higher depressive symptomatology [16]. A Japanese version of the 15-item GDS has been evaluated for validity and reliability in a Japanese population, with the recommended optimal cut-off score of 6/7 (an optimal cutoff point of 6/7 had a sensitivity of 0.98 and specificity of 0.86 while Cronbach’s alpha reliability coefficient was 0.83) [17]. We applied this recommendation in our study to define depressive and non-depressive symptoms. Participants with a GDS score of 0–6 were categorized as individuals without depressive symptoms while those with a GDS score of 7 or higher were categorized as having depressive symptoms. In our analysis, we selected only participants that answered at least 13 items.

### 2.3. Nutrients Assessment

Data related to dietary intake, including the consumption of the nine minerals of potassium, calcium, magnesium, phosphorus, iron, zinc, copper and manganese, were assessed using a brief-type self-administered diet history questionnaire (BDHQ). This was a validated BDHQ, which was designed to ask about the consumption frequency of a selected food. This used specified serving sizes described in terms of the natural portion or standard weight and volume measurement of servings. We used this BDHQ to estimate the dietary intake of 58 food and beverage items during the previous month (excluding intake from dietary supplements) in the general Japanese population [18,19,20]. These were listed as the most commonly types of food and beverages consumed in Japan as suggested by the National Health and Nutrition Survey of Japan [21]. The BDHQ consists of intake frequency of food and beverage items, daily intake of rice and miso soup, frequency of drinking and amount per drink for alcoholic beverages, usual cooking method, and general dietary behavior. Nutrient intake was calculated using an ad-hoc computer algorithm based on the Japanese standard of food composition table [22] which included weighting factors for the BDHQ [20]. Focusing on the validity of the BDHQ using 16-day weighed dietary records as the standard, it has been shown that the correlation coefficients were >0.50 for intake of all types of minerals in both genders used in our study (except for the correlation coefficient of zinc in female participants, which was 0.38) [23]. We also adjusted the basic index from the BDHQ, including total energy, protein, lipid, and carbohydrate intake. We excluded all participants who reported a total energy intake of less than 600 kcal/day (half of the required energy for the lowest physical activity category) or more than 4000 kcal/day (1.5 times the energy intake required for the moderate physical activity category) due to under- and over-estimations leading to bias in the analysis of other nutrients [19].

### 2.4. Other Variables

Demographic characteristics, including age, gender, living status (living alone or with someone), having a job (yes/no), marital status (single, got married, divorced/separated), and smoking status (current-smoker, ex-smoker, non-smoker), were obtained. Body mass index (BMI) was calculated by current body weight (in kg) divided by the squared value of body height (in meters). Participants were asked whether they had a history of hypertension, diabetes, and hyperlipidemia (self-reported physician diagnosis). Alcohol intake (g/1000 kcal) was analyzed based on the BDHQ questionnaire. 

### 2.5. Statistical Analysis

Mineral intake was adjusted for energy using the density method as a percentage of the daily energy intake for energy-containing nutrients. The distribution of variables was checked by the Kolmogorov–Smirnov and Shapiro–Wilk normality test before using other statistic tests. Student’s *t*-tests and the chi-square test were used to compare the differences in the mean level of continuous variables and categorized variables between participants with and without depressive symptoms. To examine the interaction in mineral intake between the depressive symptoms group and gender, a two-way analysis of variance (two-way ANOVA) was applied. A logistic regression analysis adjusted for potential confounding factors (age, BMI, living status, having job, married status, smoking status, alcohol consumption, energy and a history of hypertension, diabetes, and hyperlipidemia) was conducted to examine the association between mineral intake and depressive symptoms in all participants and in each gender. Regarding mineral intake, both continuous variables and variables categorized into quartiles (Appendix A for each gender), with the lowest quartile category as a reference (quartile 1: <25th percentile, quartile 2: 25–50th percentile, quartile 3: 50–75th percentile, and quartile 4: ≥75th percentile) were used in the analysis. 

Data were statistically analyzed using the Statistical Package for Social Sciences (SPSS) software program for Microsoft Windows, version 24.0 (SPSS, Inc., New York, NY, USA). Two-sided *p*-values of <0.05 were considered to represent statistically significant differences for all analyses. 

## 3. Results

### 3.1. Characteristics of Participants in Non-Depressive and Depressive Symptoms Groups.

The characteristics of the analyzed sample population according to depressive symptoms are shown in Table 1. Of the 1423 participants, 280 (20%) individuals had depressive symptoms. Subjects with depressive symptoms were significantly older (77.2 years old) than participants without depressive symptoms (73.5 years old). The level of nutrition intake was significantly lower in participants with depressive symptoms. The consumption of minerals in depressed participants was lower than in non-depressed participants, except for sodium (*p* = 0.231) and manganese (*p* = 0.417). 

### 3.2. Mineral Intake in Non-Depressive and Depressive Symptoms Groups in Each Gender

There was no difference in the mineral intake of male participants with and without depressive symptoms. In contrast, among female participants, with the exception of sodium and manganese, the intake of all other minerals was significantly lower in participants with depressive symptoms compared to those without such symptoms (Table 2). There was a significant interaction between gender and the depressive symptom group in terms of the total intake of minerals, except for sodium and manganese. This result suggests that gender plays an important role in the relationship between the intake of minerals and depressive symptoms.

### 3.3. Relationship between Mineral Intake and Depressive Symptoms 

Our results have shown that mineral intake was correlated significantly with depressive symptoms (Appendix A). To investigate the impact of gender, we clarified the relationship between mineral intake and depressive symptoms in male and female participants (Table 3). In male participants, no significant correlation between any mineral intake and depressive symptoms was found. In contrast, seven out of nine minerals showed a significant negative correlation with depressive symptoms in female participants; including potassium, calcium, magnesium, phosphorus, iron, zinc, and copper with odd ratios (OR) with 95% confidence interval (CI) of 0.473 (0.297–0.753), 0.998 (0.998–1.000), 0.990 (0.985–0.996), 0.998 (0.997–1.000), 0.802 (0.676–0.952), 0.731 (0.541–0.987), and 0.060 (0.009–0.386), respectively.

We emphasized the findings by exploring the relationship between levels of mineral intake and depression in different genders (Table 4). In male participants, no significant correlation between mineral intake and depressive symptoms was found. In contrast, six out of nine minerals showed a significant negative correlation with depressive symptoms in female participants, including potassium, calcium, magnesium, phosphorus, zinc, and copper with ORs (95% CI) from the highest to lowest quartile of 0.346 (0.191–0.627), 0.354 (0.198–0.633), 0.382 (0.215–0.677), 0.436 (0.243–0.780), 0.415 (0.231–0.746), and 0.403 (0.232–0.698), respectively. Although iron did not show a clearly significant association with depressive symptoms (*p* = 0.050), a significant inverse correlation of iron intake in the highest quartile versus lowest quartile was observed with an OR (95%) of 0.484 (0.279–0.841).

## 4. Discussion

The present cross-sectional study suggested that mineral intake, including potassium, magnesium, iron, zinc, and copper, was inversely correlated with the prevalence of depressive symptoms among Japanese elderly individuals. In particular, a strong negative association was observed between depressive symptoms and potassium, calcium, magnesium, phosphorus, iron, zinc, and copper intake in female participants, but not male participants. To the best of our knowledge, this is the first study that investigated the relationship between mineral intake and depressive symptoms in the elderly population, which was further stratified according to gender.

The association of potassium, magnesium, iron, zinc and copper intake with depression has been investigated in previous studies. Epidemiological studies on potassium intake and its relationship with depression are limited. We found only one cross-sectional study that was similar to ours, which had shown a significantly lower amount of potassium intake in a group with depression [24]. Regarding magnesium intake, the association between magnesium intake and depressive symptoms is inconsistent among studies. This present study confirmed previous findings of an inverse association between magnesium intake and depressive symptoms [25,26,27]. However, other studies did not find an association between magnesium intake and depressive symptoms [28,29]. Regarding zinc intake, our results showed that a higher level of zinc intake might be related to a lower prevalence of depressive symptoms. This finding is consistent with findings from several studies, including cross-sectional studies [10,30] and a longitudinal study [31]. There were variable findings from studies focusing on a possible association between iron intake and depressive symptoms. Our present findings are consistent with the findings that indicated a significant association between iron intake and depression [9,10]. However, another study found no significant association [32]. With regard to copper intake, to our knowledge, there is a lack of studies that have explored the association between copper intake and depressive symptoms. An inverse association between copper intake and depressive symptoms in our results is similar to the findings of one case-control study in Korean adolescent girls [33] and the finding of a cross-sectional study on adults in the US [9]. Regarding calcium intake, the study by Miki that was conducted in Japanese employees suggested that a higher dietary intake of calcium was associated with a lower prevalence of depressive symptoms [10]. Although our findings contrasted with their results, we have shown a similar association in female participants only. Our present results are also consistent with the findings in previous studies that investigated only women [34,35]. Regarding phosphorus intake, only one study examined the relationship between phosphorus intake and depression in women. However, they reported no significant association between phosphorus intake and depression [33]. Kaner showed a lower amount of phosphorus intake in the group with depressive symptoms compared to the group without such symptoms [24]. This present study is the first to indicate a negative association between phosphorus intake and depressive symptoms in women. The difference in the target population between our present study and other studies can limit appropriate comparisons with our results to different genders. 

The mechanism explaining the impact of minerals on depressive symptoms is still unclear, but some of them have been suggested. Zinc has been shown to influence brain-derived neurotrophic factor (BDNF) activities, which were found to be related to depression [36], and to affect depression by reducing several makers of inflammation, such as C-reactive protein or interleukin-6 [37]. Furthermore, zinc has antioxidant properties that may explain the pathophysiology of depression through an oxidative stress mechanism [38]. In rat models, iron was identified to play a role in oxygenation of brain parenchyma and the synthesis of neurotransmitters including dopamine and serotonin [39,40]. Magnesium was suggested to be associated with depression via its role as a protector of the nervous system [41]. Magnesium may be related to depression not only through its strong anti-inflammatory effects [42] but also through its effect on the function of the hypothalamic–pituitary–adrenal axis, which can change the level of stress hormones such as catecholamines and cortisol, thus affecting depressive symptoms [43]. Zinc, iron, and magnesium have been suggested to be related to the activation of N-methyl-D-aspartate (NMDA) receptors involved in depression [44,45,46]. A previous study investigated the impact of calcium on the synthesis of serotonin [47], which could be part of a pathway of depression [48]. Moreover, calcium works as a signal in the cells of the immune system [45], and the change in extracellular calcium concentration may influence the excitability of neuromuscular tissues involved in emotional regulation [49]. Regarding copper, a study by Jones explored how copper structurally alters serotonin and this process may play a role in copper-related neurodegenerative diseases [50]. In addition, abnormal interactions of iron or copper with metal-binding proteins that lead to oxidative stress are suggested as important mechanisms in brain aging and neurodegenerative disorders [51]. Furthermore, dietary salt intake and potassium supplementation were found to be related to dopamine levels [52], which may be a potential mechanism of depression [53,54]. The roles of phosphorus in depression are not well established and future studies are needed to explore the underlying mechanisms.

Our present study indicated that the association between mineral intake and depressive symptoms was found only in women, but not men. So far, only one study by Maserejian, which was conducted in the general population, identified that low levels of dietary or supplemental zinc are associated with depressive symptoms among women, but not men [55]; however, the mechanism causing this different has not been clearly elucidated. The higher prevalence of depression in women compared to men has been confirmed by many studies [56,57,58], and the gender difference was indicated as an important factor that affects clinical manifestations, treatment response, and control of depression [59,60,61]. Gender differences in both neurostructural and neurofunctional parameters were suggested as possible factors that might be associated with depressive symptoms of which gender differences in some serotonergic systems might play a role in the pathophysiology of depression [62]. Furthermore, gender differences exist in the genetic contributions of the serotonin transporter in depression [63], and the process of some serotonin systems might be more apparent in women than men [64]. Taken together, the gender differences in the serotonin system and the relationship between minerals and serotonin systems may explain the results found in female paticipants in our present study.

This current study has several strengths and limitations. This study was focused on only elderly people who were 65 years old or older and had a large sample size. Furthermore, we conducted a gender-stratified sub-analysis. We also took the highest cut-off point of GDS for defining depressive symptoms to strengthen the criteria used in our study. Additionally, this study identified an inverse association between mineral intake and depressive symptoms even after adjustment for potential confounders. Nonetheless, the cross-sectional design of this study cannot allow for causal inference. BDHQ data-collecting relied on the self-reporting of respondents, which imposes limitations related to the dietary assessment method. Our study focused only on minerals but not on other nutrients, such as the group B vitamins or fatty acids, which have been suggested to be related to depression in previous studies. Therefore, we cannot rule out whether the impact of minerals on the prevalence of depressive symptoms occurs independently of other nutrients. For example, an interaction between zinc and omega-3 fatty acid (particularly with docosahexaenoic acid—DHA) in human neuronal cells has been previously determined [65]. Moreover, since we used GDS to determine the depressive symptoms of participants, this only captured elevated depressive symptoms rather than clinically diagnosed depressive disorders; furthermore, as we did not correct the analyses for multiple comparisons, we cannot negate this limitation to our results. Likewise, even using various potential confounders in the analysis, there was a lack of information on other variables, such as the use of supplemental minerals, physical activity, economic income, a history of drug use, and a history of related diseases.

In conclusion, our results indicate that there is an association between mineral intake deficiencies and depressive symptoms in Japanese elderly people, particularly in female participants. Although a Japanese diet was suggested as one of the key recommendations for the prevention of depression [66], given the current findings, a preventive diet program should take into account mineral intake among elderly participants, especially women. Future, larger prospective cohort studies using serum mineral in blood for both genders are needed to verify and confirm these present findings.

## Figures and Tables

**Figure 1 nutrients-11-00389-f001:**
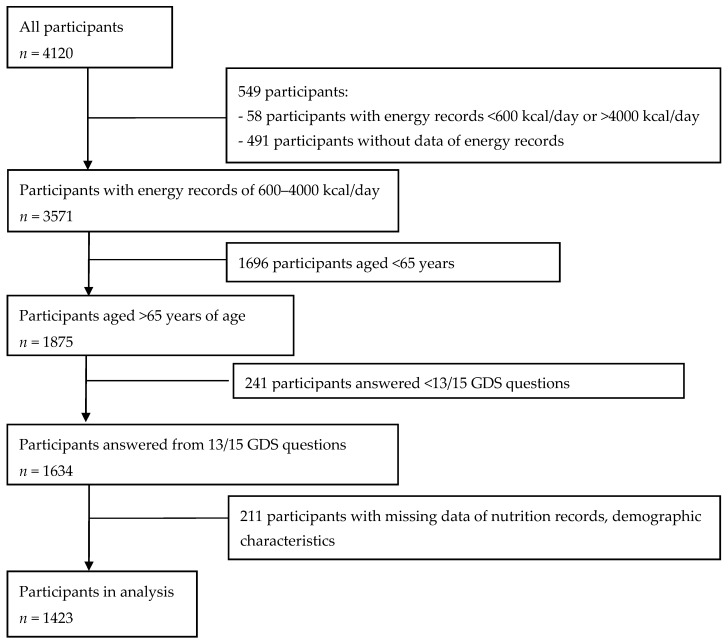
Flow chart of participant recruitment.

**Table 1 nutrients-11-00389-t001:** Comparison of characteristics among participants with and without depressive symptoms.

Characteristics		Non-Depressive Symptoms (1143, 80%)	Depressive Symptoms (280, 20%)	*p*-Value
Age (mean ± SD)		73.5 ± 7.0	77.2 ± 8.3	<0.001
Gender (n, %)	Male	522 (45.7)	131 (46.8)	0.737
	Female	621 (54.3)	149 (53.2)	
Drinking alcohol (*n*, %)	Yes	697 (61.0)	144 (51.4)	0.004
	No	446 (39.0)	136 (48.6)	
Smoking status (*n*, %)	Current-smoker	116 (10.2)	26 (9.3)	0.282
	Ex-smoker	309 (27.0)	89 (31.8)	
	Non-smoker	718 (62.8)	165 (58.9)	
Living status (*n*, %)	Alone	139 (12.2)	41 (14.6)	0.263
	With other	1004 (87.8)	239 (85.4)	
Married status (*n*, %)	Single	25 (2.2)	41 (14.6)	0.001
	Married	843 (73.8)	239 (85.4)	
	Divorce/Die	275 (24.1)	6 (2.1)	
Having a job (*n*, %)	Yes	280 (24.5)	35 (12.5)	<0.001
	No	863 (75.5)	245 (87.5)	
Hypertension (*n*, %)	Yes	463 (40.5)	114 (40.7)	0.950
	No	680 (59.5)	166 (59.3)	
Diabetes (*n*, %)	Yes	148 (12.9)	38 (13.6)	0.782
	No	995 (87.1)	242 (86.4)	
Hyperlipidemia (*n*, %)	Yes	227 (19.9)	50 (17.9)	0.448
	No	916 (80.1)	230 (81.2)	
BMI (mean ± SD)		23.0 ± 3.0	22.6 ± 3.6	0.057
Alcohol consumption (mean ± SD)		4.48 ± 8.21	3.54 ± 8.03	0.082
Total energy (kcal/day) (mean ± SD)		1862.96 ± 615.78	1735.79 ± 560.58	0.002
Nutrient intake (g/day) (mean ± SD)				
Protein		73.97 ± 30.99	66.96 ± 27.38	0.001
Animal protein		43.22 ± 24.06	38.38 ± 21.56	0.001
Vegetable protein		30.75 ± 10.67	28.58 ± 9.81	0.001
Lipid		51.69 ± 21.25	47.24 ± 20.01	0.002
Animal lipid		24.72 ± 12.61	22.00 ± 11.40	<0.001
Vegetable lipid		26.96 ± 11.27	25.24 ± 11.38	0.022
Carbohydrates		253.44 ± 85.60	243.18 ± 79.94	0.058
Ash		19.87 ± 7.22	17.67 ± 6.06	<0.001
Mineral intake (mean ± SD)				
Sodium (g/1000 kcal)		2.50 ± 0.55	2.46 ± 0.55	0.231
Potassium (g/1000 kcal)		1.47 ± 0.44	1.35 ± 0.40	0.001
Calcium (mg/1000 kcal)		327.56 ± 116.17	308.60 ± 117.27	0.016
Magnesium (mg/1000 kcal)		146.34 ± 34.50	136.85 ± 31.96	0.001
Phosphorus (mg/1000 kcal)		603.40 ± 136.54	582.05 ± 134.34	0.018
Iron (mg/1000 kcal)		4.46 ± 1.20	4.19 ± 1.04	0.001
Zinc (mg/1000 kcal)		4.54 ± 0.67	4.43 ± 0.64	0.010
Copper (mg/1000 kcal)		0.66 ± 0.11	0.65 ± 0.11	0.013
Manganese (mg/1000 kcal)		1.68 ± 0.49	1.66 ± 0.50	0.417

BMI: body mass index; SD: standard deviation.

**Table 2 nutrients-11-00389-t002:** Difference of minerals intakes between depressive and none-depressive symptoms in each gender.

Minerals Intakes (Mean ± SD)	Male			Female			*p* *
	Non-Depressive Symptoms (*n* = 522, 79.9%)	Depressive Symptoms (*n* = 131, 29.1%)	*p*-Value	Non-Depressive Symptoms (*n* = 621, 80.6%)	Depressive Symptoms (*n* = 149, 19.4%)	*p*-Value	
Sodium (g/1000 kcal)	2.49 ± 0.57	2.47 ± 0.56	0.745	2.51 ± 0.53	2.45 ± 0.55	0.173	0.511
Potassium (g/1000 kcal)	1.35 ± 0.38	1.32 ± 0.39	0.443	1.57 ± 0.46	1.38 ± 0.41	<0.001	0.003
Calcium (mg/1000 kcal)	299.77 ± 99.10	308.61 ± 121.44	0.442	350.93 ± 124.13	308.59 ± 113.88	<0.001	0.001
Magnesium (mg/1000 kcal)	139.13 ± 30.01	136.94 ± 31.66	0.461	152.40 ± 36.80	136.76 ± 32.32	<0.001	0.003
Phosphorus (mg/1000 kcal)	574.38 ± 122.65	584.48 ± 137.45	0.411	627.80 ± 142.81	579.93 ± 131.97	<0.001	0.001
Iron (mg/1000 kcal)	4.24 ± 1.11	4.14 ± 1.04	0.342	4.64 ± 1.24	4.22 ± 1.04	<0.001	0.045
Zinc (mg/1000 kcal)	4.39 ± 0.65	4.39 ± 0.63	0.931	4.67 ± 0.66	4.46 ± 0.66	0.001	0.014
Copper (mg/1000 kcal)	0.64 ± 0.11	0.64 ± 0.11	0.889	0.69 ± 0.11	0.65 ± 0.11	<0.001	0.012
Manganese (mg/1000 kcal)	1.66 ± 0.51	1.66 ± 0.48	0.993	1.70 ± 0.47	1.65 ± 0.51	0.259	0.455

*p*-value: Comparison of mineral intake between groups of depressive symptoms and non-depressive symptom in each gender using independent *t*-test. *p* *: Interaction between gender and depressive symptoms by minerals intake (two-way analysis of variance (ANOVA)).

**Table 3 nutrients-11-00389-t003:** Odd ratios and 95% CI for depressive symptoms across mineral intake stratified by gender ^¶^.

Minerals	Male			Female		
	B	OR (95% CI)	*p*-Value	B	OR (95% CI)	*p*-Value
Sodium	−0.316	0.729 (0.502–1.058)	0.096	−0.187	1.030 (0.969–1.195)	0.315
Potassium	−0.343	0.710 (0.408–1.234)	0.225	−0.750	0.473 (0.297–0.753)	0.002
Calcium	<0.001	1.000 (0.098–1.002)	0.743	−0.002	0.998 (0.996–1000)	0.016
Magnesium	−0.003	0.997 (0.990–1.003)	0.326	−0.010	0.990 (0.985–0.996)	0.001
Phosphorus	<0.001	1.000 (0.999–1.002)	0.657	−0.002	0.998 (0.997–1.000)	0.022
Iron	−0.163	0.850 (0.702–1.029)	0.095	−0.220	0.802 (0.676–0.952)	0.012
Zinc	−0.221	0.802 (0.562–1.144)	0.223	−0.313	0.731 (0.541–0.987)	0.041
Copper	−1.266	0.282 (0.037–2.165)	0.224	−2.813	0.060 (0.009–0.386)	0.003
Manganese	−0.309	0.734 (0.484–1.114)	0.146	−0.225	0.799 (0.530–1.205)	0.284

B: coefficient, OR: odd ratio, CI: confidence interval. ^¶^ adjusted for age, BMI, living status, having a job status, married status, smoking status, drinking alcohol, alcohol consumption, total energy, hypertension, diabetes, hyperlipidemia.

**Table 4 nutrients-11-00389-t004:** Odd ratios and 95% CI for depressive symptoms across level of mineral intake stratified by gender ^¶^.

Minerals		Male		Female	
		B	OR (95%CI)	B	OR (95%CI)
Sodium	Quartile 1		1.000 (reference		1.000 (reference)
	Quartile 2	−0.421	0.656 (0.364–1.184)	−0.631	0.532 (0.309–0.916) *
	Quartile 3	−0.133	0.876 (0.501–1.529)	−0.334	0.716 (0.426–1.204)
	Quartile 4	−0.475	0.622 (0.342–1.131)	−0.165	0.848 (0.506–1.423)
	*p*-value	0.154		0.118	
Potassium	Quartile 1		1.000 (reference)		1.000 (reference)
	Quartile 2	0.161	1.175 (0.671–2.058)	−0.053	0.948 (0.580–1.552)
	Quartile 3	−0.519	0.595 (0.323–1.094)	−0.324	0.724 (0.428–1.222)
	Quartile 4	−0.188	0.829 (0.465–1.477)	−0.911	0.402 (0.220–0.737) *
	*p*-value	0.132		0.004	
Calcium	Quartile 1		1.000 (reference)		1.000 (reference)
	Quartile 2	−0.033	0.967 (0.551–1.698)	−0.382	0.683 (0.413–1.129)
	Quartile 3	−0.713	0.490 (0.261–0.921) *	−0.306	0.736 (0.443–1.224)
	Quartile 4	0.106	1.111 (0.643–1.920)	−1.007	0.365 (0.203–0.658) *
	*p*-value	0.082		0.006	
Magnesium	Quartile 1		1.000 (reference)		1.000 (reference)
	Quartile 2	0.037	1.038 (0.594–1.811)	−0.082	0.921 (0.564–1.506)
	Quartile 3	−0.256	0.774 (0.434–1.377)	−0.368	0.692 (0.409–1.170)
	Quartile 4	−0.247	0.781 (0.443–1.378)	−0.891	0.410 (0.226–0.744) *
	*p*-value	0.517		0.010	
Phosphorus	Quartile 1		1.000 (reference)		1.000 (reference)
	Quartile 2	0.053	1.055 (0.594–1.872)	−0.118	0.889 (0.540–1.463)
	Quartile 3	−0.346	0.707 (0.385–1.299)	−0.254	0.776 (0.465–1.296)
	Quartile 4	0.208	1.231 (0.701–2.165)	−0.818	0.441 (0.243–0.803) *
	*p*-value	0.426		0.034	
Iron	Quartile 1		1.000 (reference)		1.000 (reference)
	Quartile 2	−0.152	0.859 (0.485–1.523)	−0.115	0.891 (0.534–1.487)
	Quartile 3	−0.372	0.689 (0.381–1.246)	−0.120	0.887 (0.555–1.418)
	Quartile 4	−0.212	0.809 (0.458–1.428)	−1.192	0.304 (0.101–0.917) *
	*p*-value	0.366		0.050	
Zinc	Quartile 1		1.000 (reference)		1.000 (reference)
	Quartile 2	0.109	1.115 (0.613–2.030)	−0.082	0.922 (0.558–1.522)
	Quartile 3	−0.054	0.948 (0.509–1.765)	−0.126	0.882 (0.532–1.461)
	Quartile 4	−0.187	0.830 (0.444–1.550)	−0.771	0.463 (0.255–0.839) *
	*p*-value	0.584		0.023	
Copper	Quartile 1		1.000 (reference)		1.000 (reference)
	Quartile 2	−0.255	0.775 (0.422–1.424)	−0.209	0.811 (0.493–1.335)
	Quartile 3	−0.253	0.776 (0.426–1.414)	−0.497	0.608 (0.361–1.024)
	Quartile 4	−0.255	0.775 (0.421–1.424)	−0.837	0.433 (0.245–0.766) *
	*p*-value	0.541		0.005	
Manganese	Quartile 1		1.000 (reference)		1.000 (reference)
	Quartile 2	0.188	1.207 (0.670–2.175)	0.080	1.084 (0.645–1.820)
	Quartile 3	0.413	1.511 (0.843–2.707)	−0.224	0.800 (0.467–1.370)
	Quartile 4	−0.190	0.827 (0.448–1.528)	−0.291	0.747 (0.430–1.299)
	*p*-value	0.460		0.361	

B: coefficient, OR: odd ratio, CI: confidence interval. ^¶^ adjusted for age, BMI, living status, having a job status, married status, smoking status, alcohol consumption, total energy, hypertension, diabetes, hyperlipidemia; * *p* < 0.05.

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
