# Peer review of "Association between Lower Intake of Minerals and Depressive Symptoms among Elderly Japanese Women but Not Men: Findings from Shika Study"

_nutrients, 2019, doi:10.3390/nu11020389_

Round 1
Reviewer 1 Report
General comment
In this cross-sectional study, the authors examined the association between mineral intake and depressive symptoms in 1423 Japanese men and women aged > 65 years. The authors have found that lower dietary intake of potassium, calcium, magnesium, phosphorus, iron, zinc and copper were associated with higher depressive symptoms in women. No associations between minerals intake and depressive symptoms were found in men. Despite the importance of the research question, this study presents several weaknesses. Mineral intake assessment was performed using a food frequency questionnaire which does not assess portion size, a 24h food diary recall would have been more appropriate. Further, there were no assessment of dietary supplement by the participants. In addition, the authors did not correct for multiple comparisons and did not discuss it as a potential limitation. Importantly, the writing style has to be improved, the manuscript should be reviewed by a native English speaker. The authors should also check the format of the author citation in the text.
Specific comments:
Introduction
The introduction of the topic could be improved. Please introduce why mineral intake may be associated with depressive symptoms. Please explain why it is relevant to study both men and women separately and the elderly population specifically.
Method
Why use a food frequency questionnaire and not a 24 hour recall with portion sizes?
Can you provide some references of studies having validated this method to assess mineral intake?
Statistics: did you correct for multiple comparisons?
Discussion
Line 182 “However, other studies identified no association between magnesium intake and depressive symptoms [24], [25].” Did you mean did not find an association?
Minerals intake co-occurs with intake of other nutrients that could potentially also affect mood such as Omega-3 fatty acids in seafood. Please discuss this point.
The section on the biological mechanisms explaining the associations between mineral intake and depression could be developed more.
Please add to the limitation that you did not correct your analysis for multiple comparisons.
Author Response
Dear Reviewer,
Thank you very much for giving us valuable comments about our manuscript. According to your suggestion, we have changed and added more information to the new version of our manuscript. We do hope the new manuscript will satisfy you.
For details, we would like to explain and answer your specific comments:
-Introduction:
According to your advice, we have improved our manuscript by adding more information between mineral intake and depression.
-Method:
+ We are very sorry about our mistakes in writing the paragraph of using BDHQ. We wrote that “BDHQ was designed to ask about the consumption frequency of selected food, but not portion size…”. Thanks to your comment, we found our mistake. Actually, BDHQ was designed to ask about the consumption frequency of selected food, with specified serving sizes described in terms of the natural portion or standard weight and volume measurement of servings. We changed that line in the revised version.
In Japan, BDHQ is a famous questionnaire for using to estimate the dietary intake in general Japanese populations. This questionnaire was developed and validated and is used commonly. We cannot deny that there are still limitations of using this questionnaire. However, this questionnaire was developed based on the eating habit of Japanese people (food consumption, cooking style…). Besides that, our study was conducted in a rural area with a very traditional eating habit. Therefore, we think that this is the most suitable questionnaire for using in our study.
We cited one more reference of BDHQ of S. Kobayashi et al., “Both comprehensive and brief self-administered diet history questionnaires satisfactorily rank nutrient intakes in Japanese adults”. This study validated BDHQ to assess mineral intake. A recent paper of Miki et al also used BDHQ to access mineral intake "Longitudinal adherence to a dietary pattern and risk of depressive symptoms: the Furukawa Nutrition and Health Study. Nutrition. 2018 Apr 1 Available from: https://www.sciencedirect.com/science/article/pii/S0899900717302575
+ We did not correct for multiple comparisons in analysis and added it in the limitation of our study.
-Discussion
+ Thank you for your comment, we changed the way of writing in line 182 to make it clearer, from original sentence “However, other studies identified no association between magnesium intake and depressive symptoms” to “However, other studies did not find an association between magnesium intake and depressive symptoms”. In the revised version, it goes to line 247.
+ We understand that together with minerals, other nutrients intake such as vitamins group B or fatty acids could also affect mood. Besides that, many studies on the association between macronutrients and depression are also being conducted. Therefore, we cannot deny the cumulative impact of nutrients and their interactions on depression. According to your suggestion, we added one other limitation of our study about the co-occurrence between nutrients.
+ We have discussed more the biological mechanism explaining the association between mineral intake and depression.
For the details of the change, please see the revised version that we did.
Once again, we would like to express our sincere thanks to you for your contribution. We do hope the change that we made will satisfy you.
The authors

Reviewer 2 Report
The subject is very interesting and new data are given, but the results are not fully presented. A clear description of results, correlations among them and comparison respect to previous work should be given.
Also a more efficace title should be given.
The Introduction should be rewritten. Now it is poorly formulated.
A graphical scheme of design of study should be inserted.
The advantage and innovative side of this work should be inserted.
Author Response
Dear Reviewer,
Thank you very much for giving us valuable comments about our manuscript. According to your suggestion, we changed and added more information to a new version of our manuscript:
-About the results part, first of all, we performed the characteristics of participants with mineral intake by groups with depressive symptoms and without depressive symptoms. Then we compared mineral intake between 2 groups in both genders. After that, we used logistic regression (both with continuous variables and categorized variables) to perform the association between mineral intake and depressive symptoms in males and females. Besides that, we made supplementary to show more details of our results. We changed the title of table 1 to make it clearer.
-According to your advice, we added more information to explain our purpose of this study in the introduction part.
- We also created a graphical scheme of the design of our study in the method part.
-The advantage side of this result was added in the part of the conclusion.
Please see the details of the change in the revised version.
Once again, we would like to express our sincere thanks to you for your contribution. We do hope the change that we made will satisfy you.
The authors

Round 2
Reviewer 2 Report
The authors have improve the manuscript.
Author Response
Dear Reviewer,
Thank you very much for your comment!
The Authors